# Biomarkers for Alzheimer’s Disease in the Current State: A Narrative Review

**DOI:** 10.3390/ijms23094962

**Published:** 2022-04-29

**Authors:** Serafettin Gunes, Yumi Aizawa, Takuma Sugashi, Masahiro Sugimoto, Pedro Pereira Rodrigues

**Affiliations:** 1Ph.D. Program in Health Data Science, Faculty of Medicine, University of Porto, 4200-450 Porto, Portugal; toroscm@gmail.com; 2Center for Minimally Invasive Therapies, Institute of Medical Science Research and Development, Tokyo Medical University, Tokyo 160-8402, Japan; azyumi726@gmail.com (Y.A.); oshou.0131@gmail.com (T.S.); 3RISE & MEDCIDS, Faculty of Medicine, University of Porto, 4200-450 Porto, Portugal; pprodrigues@med.up.pt

**Keywords:** Alzheimer’s disease (AD), biomarkers, low or non-invasively, machine-learning classification

## Abstract

Alzheimer’s disease (AD) has become a problem, owing to its high prevalence in an aging society with no treatment available after onset. However, early diagnosis is essential for preventive intervention to delay disease onset due to its slow progression. The current AD diagnostic methods are typically invasive and expensive, limiting their potential for widespread use. Thus, the development of biomarkers in available biofluids, such as blood, urine, and saliva, which enables low or non-invasive, reasonable, and objective evaluation of AD status, is an urgent task. Here, we reviewed studies that examined biomarker candidates for the early detection of AD. Some of the candidates showed potential biomarkers, but further validation studies are needed. We also reviewed studies for non-invasive biomarkers of AD. Given the complexity of the AD continuum, multiple biomarkers with machine-learning-classification methods have been recently used to enhance diagnostic accuracy and characterize individual AD phenotypes. Artificial intelligence and new body fluid-based biomarkers, in combination with other risk factors, will provide a novel solution that may revolutionize the early diagnosis of AD.

## 1. Introduction

Dementia is one of the leading health concerns in aging societies [1,2]. The World Health Organization suggests that approximately 10 million new cases of dementia occur every year worldwide, and approximately 60–70% of these are patients with Alzheimer’s disease (AD) [3]. Due to aging trends, especially in industrialized countries, the burden of AD in the next decade will be very high, impacting hundreds of millions of people, their families, and the national health care systems [4]. Epidemiological evidence suggests that in the United States, one in every three individuals aged over 85 years will develop AD [5], and by 2050, the number of Americans aged over 85 years will triple [6]. AD is among the five leading causes of death in industrialized countries. Furthermore, it is the only disease forecasted to grow by a large enough margin to become the primary disease in the following decades [7]. Therefore, reversing these trends is critical worldwide.

Effective treatment to cure AD is not currently available [8]. Once AD is diagnosed, the disease shows irreversible progression, although there are differences in progression among individuals [9]. AD aggravation affects patients and their lifetime [10]. The survival rate of AD varies from three to nine years, depending on the age at symptom onset, which puts it in the group of long-term conditions [11]. Although a few patients suffer from juvenile AD (diagnosed < 40 years old), this review focuses on patients with late-onset Alzheimer’s disease (LOAD).

AD in the older population starts developing a decade or more before the actual diagnosis [12]. Patients with AD first go through the mild cognitive impairment (MCI) phase, which can continue for several years. MCI is also split into the early and late MCI stage characteristics. Not all MCI phases progress to AD [13]. Many of the risk factors for LOAD were already identified (e.g., high cholesterol, type II diabetes, high blood pressure, obstructive sleep apnea, and classic socioeconomic determinants, such as low education) [14]. Early AD diagnosis can improve the prognosis [15]; however, the current gold standard of AD diagnosis is the pathological analysis of brain tissue, an option not feasible for mass screening owing to its invasiveness [16]. Thus, less invasive methods for AD diagnosis are necessary.

Brain-imaging tests/devices (i.e., different versions of positron emission tomography (PET) scans and magnetic resonance imaging (MRI) devices) are alternatives used for AD diagnostics. In addition, molecular biomarkers, such as amyloid-beta isoform 42 (Ab42) and phosphorylated tau, have been found in cerebrospinal fluid [17]. These methods, currently used in clinical settings, help identify patient status, whether they belong in the normal cognitive (NC) group, the MCI group, or the AD group. However, PET and MRI tests need expensive infrastructure [18], and CSF biomarker testing remains highly invasive and places a high physical burden on the examinee, thus limiting its use [19].

A non-invasive and diagnostic solution that can indicate the risk of developing AD with high accuracy is critically important. Having this objective in mind and taking advantage of the recent technological advances in computational power and lab equipment, many new ideas/biomarkers have been investigated and put forward by the scientific community to detect AD early before LOAD [20]. These are based on biomarkers found in the blood, plasma, serum, urine, and saliva [21,22]. However, considering the heterogeneity among patients, a single biomarker is not enough to characterize each patient. Nowadays, omics technologies enable us to profile hundreds of molecules simultaneously. Thus, the integration of quantified molecular patterns and computational power, e.g., the use of artificial intelligence (AI) and machine-learning (ML) tools, would solve these issues [23]. AI-based data analytical methods are already being studied for diagnosing AD, not only based on the biofluid biomarkers but also on other modalities (e.g., readings of the retina + iris of the eye) [24], electrical signal measurements of brain waves (electroencephalogram (EEG)) [25], and AI-based online language skills and memory tests [26]. These are promising developments that can lead to feasible mass screening of AD, as long as they have high accuracy in real clinical settings.

The use of ML and AI in numerous areas of healthcare can improve the accuracy of biomarker-based testing and tackle an even more important issue—the heterogeneity of each subject. Therefore, ML and AI can open promising avenues for accurate, non-invasive, and accessible early diagnosis of AD and support subject-specific prognosis and treatment responses to ground personalized approaches in disease management.

This review paper describes an update on the non-invasive or minimally invasive techniques for identifying biomarkers. In addition, the topic includes AD diagnostic tests that enable mass screening to predict the risk of AD in the early and mild phases. This review also consists of the recent developments in ML employed in AD diagnosis that can be used along with the biomarkers to improve the accuracy of diagnosis and the subject differentiation capacity.

## 2. Clinical-Pathological Factors

Here we have summarized several different hypotheses and theories about the pathogenesis of AD and its causes in the elderly. AD is a neurodegenerative disease, a subgroup of a broader disease called dementia that mainly affects people aged 65 and older. Dementia is a general term for a group of diseases that impair memory, language, cognitive function, and motor function, resulting in the inability to lead a normal daily life. However, AD is the largest subgroup, accounting for 60–80% of all dementia patients. Others include Lewy body dementia, vascular dementia, and frontotemporal dementia, accounting for 5–10% of the patient population [27]. The progression of AD can be split into three major phases/groups as follows: the aggregation process of amyloid-beta (Aβ), stimulating the phosphorylation of a tau protein, and aggregation of phosphorylated tau (p-tau) in neurons. These molecules are mainly located in the hippocampus, where memory and learning (recording new knowledge) are controlled [28,29,30]. The first phase is presymptomatic, also known as NC, where there is no cognitive impairment. The second phase is MCI, where the loss of neurons in the hippocampus causes short-term memory problems. MCI may last between two to seven years. The third phase is when AD is confirmed; in this phase, both short- and long-term memory is lost, hallucinations and delusions start, and after some years, it reaches a more aggressive stage. This phase is the most difficult, and it may last anywhere between three to eight years before death occurs. In the worst phase of AD, it inflicts a tremendous mental, physical, and financial burden on caregivers (Figure 1). Therefore, it is crucial to detect AD in the NC stage before developing. In the next section, we will introduce biomarkers for detecting AD.

## 3. Conventional Biomarkers

The biomarkers are essential for the early detection of the risk of developing AD diseases before the dementia stage (Figure 2). Today, these biomarkers are clinically useful and can serve as reference validation techniques for other alternative biomarkers present in human body fluids. The recently practiced biomarkers for AD diagnosis are based on the National Institute on Aging and Alzheimer’s Association’s 2018 published research framework, which updates earlier protocols for AD diagnostic guidelines to focus on biomarkers rather than initial symptom measurement [31].

The Braak-staging classification of AD has been a widely accepted diagnostic tool for assessing the different stages of AD in other regions of the brain since the early 1990s. A recent study conducted in 2021 used the Braak stages and compared tau-based PET scan biomarkers versus Aβ-based PET scan biomarkers. It showed that the combination of a tau-PET scan and Braak staging is promising for predicting patient-specific risks of clinical AD progression compared to just using an Aβ PET scan [32,33].

### 3.1. Brain Imaging

Single-photon emission computed tomography (SPECT) is mainly used to measure blood flow and concentration degree between arterial blood and brain tissue, and it uses the application of kinetic analysis models to calculate the regional cerebral blood flow (rCBF). In addition to rCBF measurement by SPECT, dopamine transporter radiotracers are also used with SPECT and can have clinical utility in distinguishing the different forms of dementia [34].

The functional imaging (MRI, PET, and SPECT scans) tests are in particular clinically important. They can identify the subjects with a high risk of developing AD while at the MCI stage with meaningful accuracy. A meta-analysis by Yuan et al., which included 24 studies with a total of 1112 patients, found sensitivity and specificity values of 89% and 85% for PET, 84% and 70% for SPECT, and 73% and 81% for MRI, respectively, for prediction of conversion from MCI to AD [35]. These imaging tests have been successful in differentiating AD from MCI. On the other hand, PET scans show higher positivity for amyloid (A+) with increasing age, but this does not necessarily correspond to memory impairment or other cognitive conditions. Thus, visual interpretation of structural images cannot reliably identify patients with presymptomatic conditions before MCI [36].

### 3.2. High Invasive Biomarkers

The following β amyloid deposition, pathologic tau, and neurodegeneration (ATN) guidelines for AD diagnosis that combine both CSF and imaging biomarkers have been proposed by the National Institute on Aging-Alzheimer’s Association [37]. The combination of imaging and CSF biomarkers improves the accuracy of complicated AD diagnostics and prognoses. The biomarker related to Aβ in CSF is called (Aβ42) because it is the 42 long amino acid (peptide) form of Aβ in the brain tissue and is the dominant biomarker that is associated with AD disease. Its level in CSF is negatively corralled with the Aβ plaques in the brain tissue, indicating that the lower the amounts of Aβ42 in the CSF, the higher the number of Aβ plaques, as seen by PET scans. The ratio of Aβ42/Aβ40 is associated with risk progression from presymptomatic to MCI or from MCI to AD.

Tau is a protein found in the axons of the neurons in the brain. High levels of total tau and p-tau in CSF are associated with AD. An increased amount of total tau and p-tau in the CSF indicates that these are secreted from the brain tissue neurons, which can be a response to elevated Aβ plaques. P-tau stands for hyper-phosphorylated tau proteins. There are several forms of p-tau biomarkers, namely P-tau181, P-tau217, and P-tau231. The utility of each performs better in different situations, whether in the classification of AD from non-AD tau pathology or indications of progression through different AD stages. The amount of total tau in the CSF is a biomarker for the severity of neurodegeneration. Tau aggregation forms neurofibrillary tangles that damage normal neuronal plasticity and synaptic transmission processes. 

Morinaga et al. conducted a study of 207 patients with AD to examine the performance of biomarkers using MRI and PET. The study showed that AD findings were observed in 77.4% of all AD patients using MRI, 81.6% using SPECT, 93.1% using fluorodeoxyglucose (FDG) PET, and 94.0% using CSF biomarkers. At the stage of clinical dementia rating (CDR) 0.5 (questionable), sensitivity was 90.0% for CSF biomarkers, 80.8% for SPECT, 71.4% for FDG PET, and 65.5% for MRI. At CDR 1 (mild), FDG PET (96.7%) and CSF biomarkers (95.5%) were the most sensitive. All biomarkers showed high sensitivity at CDR 2 (moderate) [38]. In addition, there have been some case studies where the association between AD and these biomarkers is not confirmed [39].

These conventional currently available AD biomarker tests are either highly invasive (lumbar puncture) or expensive and labor-intensive (imaging), making them unsuitable for use in the primary care, clinical office-based setting. Therefore, the search for inexpensive biomarkers by metabolites and factors that can be collected minimally invasively from blood or urine, which solve these problems, has been underway. The next section introduces minimally invasive and non-invasive biomarkers.

## 4. Novel Non-Invasive and Minimally Invasive Biomarkers

This AD developing processes complexity needs the identification of biomarkers that enable the detection and progression of this disease. However, diagnoses using CSF biomarkers and brain imaging are invasive. Therefore, a non-invasive and accurate biomarker for the classification of AD spectrum and stages of AD is needed.

### 4.1. Non-Invasive Biomarkers

Neurons communicate and perform all functions using electrical impulses, and EEG [40] captures this electrical activity through small electrodes placed on the scalp, displaying electrical impulses as waves. Individuals with AD typically experience a general slowing of EEG, including a reduction in higher frequency waves, such as gamma. The power spectrum, complexity, and synchronization characteristics of EEG waveforms in AD patients have a distinct deviation from normal elderly individuals, indicating these EEG features can be promising candidate biomarkers of AD [41,42]. Another promising biomarker is the study of microvascular changes in the eye retina by OCT and OCTA techniques. It is based on the fact that the eyes are directly connected to the brain. A recent study of damage to the microvascular network and neural microstructure of the retina has been reported in AD, MCI, and even preclinical AD [43]. Studies of human and animal models of AD have also revealed biochemical pathways that are altered in the retina during diseases, such as Aβ and tau deposition [44].

Thus, since it is a non-invasive technique and different from most other biomarkers used by other clinical-stage biotech companies, EEG techniques with bodily fluid biomarkers may offer a more accurate prediction of AD status [45].

### 4.2. Minimally Invasive Biomarkers

Furthermore, several biomarkers are being found to predict Alzheimer’s disease from blood, saliva, and urine in addition to the CSF. For instance, thousands of proteins in blood plasma were analyzed to predict AD, and 19 hub proteins showed a predictive ability of clinical AD classification [46]. Similarly, the researchers investigated hundreds of different metabolites found in saliva to discover which ones may predict AD status. This study showed that salivary metabolite markers could discriminate between AD, presymptomatic (PP), and MCI patients; AD and PP patients were identified by the metabolite markers glucosylgalactosyl, hydroxylysine—H_2_O, and Glutamine-carnitine (discovery phase (DP) and validation phase (VP): area under the curve [AUC] = 1.000, AUC = 1.000 with 100% sensitivity and 100% specificity). The MCI and AD groups were best discriminated with metabolite markers of alanyl-phenylalanine and phenylalanyl-proline (DP: AUC = 0.779; VP: AUC = 0.889). In addition, using positively confirmed metabolites, we distinguished AD from PP and MCI with good diagnostic performance (AUC > 0.8) [47].

Furthermore, changes in gene expression specific to AD have also been noted—the AD mark Early Onset Alzheimer’s Panel. Three genes (Apo E genotype, PSEN1, and PSEN2) can detect 45–90% of early AD patients [48]. Currently, none of these tests have been approved by the U.S. Food and Drug Administration. The test is based on the precise and robust quantification of the Aβ42/40 ratio (Aβ 42/40) and ApoE genotype in blood samples. In another study, lactoferrin LF, a major antimicrobial peptide in saliva, was found in senile plaques and neurofibrillary changes in the brain. Therefore, LF expects to be a highly sensitive and specific biomarker for AD diagnosis [49]. Other enzymes, hormones, or brain-secreted exosomal contents, such as proteins, lipids, and various RNA species found in blood or saliva, can have predictive powers for detecting and classifying the AD continuum. Furthermore, it has also been reported that measurement errors can occur among institutions, even for biomarkers obtained from the same sample from the same population [50].

However, there is no standard clinical solution for non-invasive or minimally invasive early AD detection. Therefore, the scientific community is investigating analytical methods to integrate this information. The following sections introduce artificial intelligence (AI) and machine learning (ML) for the integrative data analysis.

## 5. Case for Inclusion of the ML Applications for Non-Invasive AD Diagnostics Solution

Artificial intelligence (AI) and machine learning (ML) are widely used in various fields. Deep learning is a subtype of artificial neural networks (ANN). The structure of ANN comprises input and output layers with several hidden layers. Each layer has several nodes (similar to a neuron in the brain) connecting to another node in the next layer with numerical weights. ANN-based sub-architectures have begun to be used in computer vision, such as image processing, and perceptual computing, such as signal processing. Recently, research has advanced in the field of medicine [51]. The main topics in healthcare include the following: (1) diagnosis to differentiate a specific disease from the others and (2) predicting to monitor the results of interventions.

AI and ML already have roles in the health care system, and their application and importance are heavily recognized. They have contributed positively to many areas of the delivery of health care services, including operational efficiency, personalized cancer treatment diagnostics of certain diseases and infections, and drug discovery, among many other applications. Naturally, many AI-related studies are being conducted for AD detection and prognosis. However, a large-scale database is needed to improve the accuracy of AI or ML. Many ML studies are based on several big initiatives, such as the AD Neuroimaging Initiative (ADNI) database [52]. An ML competition is also called the Alzheimer’s Disease Prediction of Longitudinal Evolution (TADPOLE) [53]. In addition to ADNI and TADPOLE initiatives, several more initiatives aim to take advantage of the latest cutting-edge application of AI-ML tools to improve AD diagnosis. One more example is the Open Access Series of Imaging Studies (OASIS) database, which consists of brain imaging data (MRI and PET scans) that have been collected at MIT from almost 1100 participants [54]. Many academic studies on AI–ML applications also use OASIS databases [55].

Medical studies tried to predict AD using the BioFinder database that collects longitudinal data, including images, CSF and plasma biomarkers, and neuropsychological tests in 1600 individuals belonging to NC, MCI, and AD groups [56]. These data are being used to develop accurate and early diagnostic methods, identify novel therapeutic targets, and elucidate the relationship between different pathologies and clinical systems [57]. Recently, a data platform called Alzheimer’s Disease Data Initiative has been collecting data from global collaborators related to AD and related dementias. Its cloud-based platform allows scientists and researchers to share and discover data to accelerate discoveries and innovations in AD and related disorders [58]. In the early stages, it classifies the subgroups that are in MCI, which may progress to AD, and those that regress to NC. More information on different AI and ML applications used in AD classification solutions [59,60]. It is more reliable to develop AI–ML models and algorithms with relevant big data to train these programs. Furthermore, it is vital to improving machine-learning models by feeding back diagnostic data to develop accurate early diagnosis techniques and new biomarkers
[61] (Figure 3). However, there are many challenges in developing the diagnosis of AD biomarkers and diagnostic devices. It is essential to promote mutual understanding with clinicians and data analysts.

## 6. Conclusions

The literature review on AD diagnosis found that both academia and biotech companies have tried to diagnose the early stage of AD, i.e., MCI. Currently, only expensive and invasive biomarkers (e.g., brain imaging, CSF Aβ 42, p-tau, and t-tau levels) are available. Considering the high heterogeneities of AD diseases, multiple biomarkers (e.g., Apo E genotype, saliva TF marker) would show a higher potential to differentiate AD patients. The current framework for diagnosing AD, which uses beta-amyloid deposition, pathological tau, and neurodegeneration (ATN) as indicators, is not fully functional. Although the currently available applications are still limited, AI–ML tools may establish a reliable solution. No standard solution is available to diagnose the complex AD continuum, especially before the MCI stage. Further research is needed to discover new minimally invasive and cost-effective biomarkers. The integrated analyses of both clinical data and multiple biomarkers would be effective methodologies for the early detection of AD patients.

## Figures and Tables

**Figure 1 ijms-23-04962-f001:**
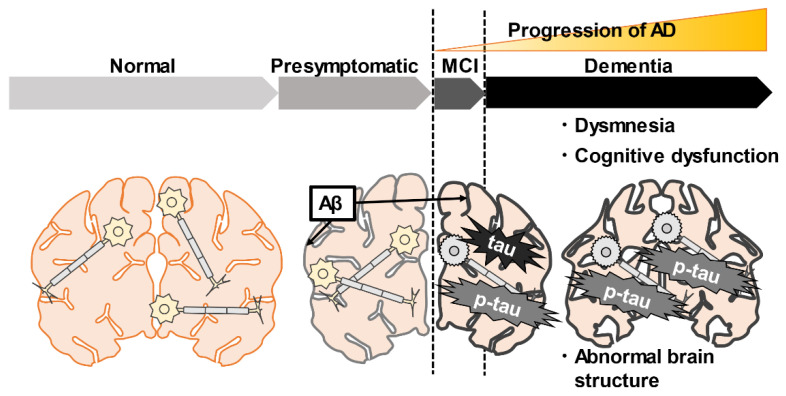
The pathogenesis and temporal changes of Alzheimer’s disease (AD). AD is characterized by the accumulation of amyloid-beta (Aβ), tau, p-tau, brain atrophy, and cognitive decline. Accumulation of Aβ occurs gradually from the presymptomatic period. In the mild cognitive impairment stage (MCI), Aβ deposits and tau-mediated neuronal damage, and short-term memory problems progress gradually. In the dementia stage, Aβ deposit and tau-mediated neuropathy further structural abnormalities of the brain and memory impairment occurs. The black lines indicate the accumulation of Aβ in the brain.

**Figure 2 ijms-23-04962-f002:**
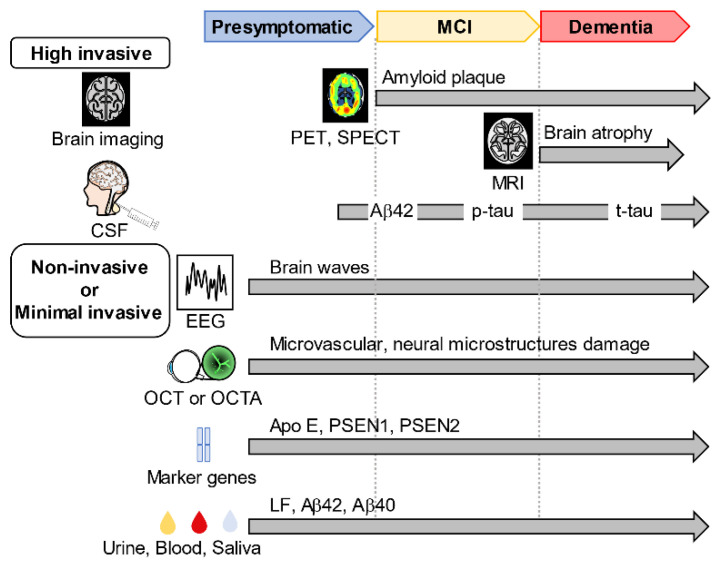
Biomarker-screening modalities for AD. Conventionally used biomarkers for Alzheimer’s disease (AD) include positron emission tomography (PET) and single-photon emission computed tomography (SPECT) to examine brain function and nuclear magnetic resonance to analyze structural changes in the brain. AD biomarkers include amyloid β 42 (Aβ42), phosphorylated tau (p-tau), and total tau (t-tau) levels in cerebrospinal fluid. New biomarkers such as optical coherence tomography (OCT) and optical coherence tomography angiography (OCTA) for measuring eye vascular abnormalities, electroencephalograms (EEG) for measuring brain waves, and AD marker genes have been found. Biomarkers in urine, blood, and saliva have also been identified.

**Figure 3 ijms-23-04962-f003:**
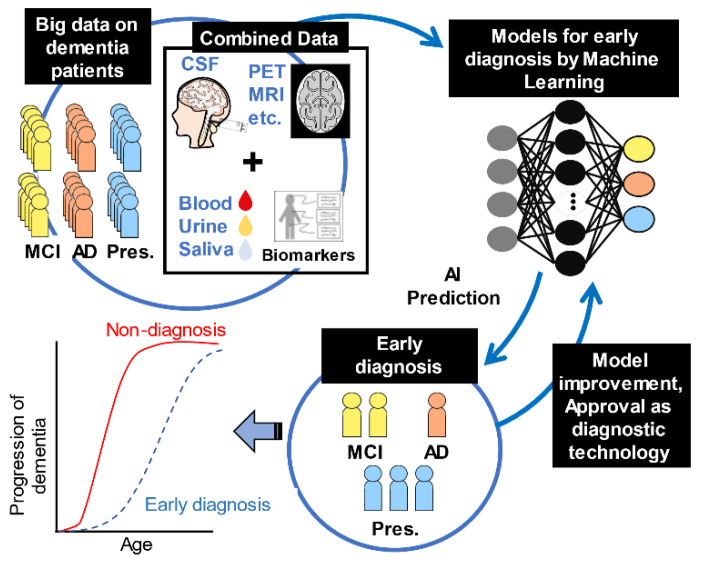
Extensive data search by machine learning. The first step is to collect big data from dementia patients. Then, they are labeled as presymptomatic (Pres.), mild cognitive impairment (MCI), and Alzheimer’s disease (AD) patients. Next, input to machine learning from a large amount of labeled data. The results of this machine learning are evaluated to calculate a model that can make accurate predictions. The next stage is the implementation phase. New unlabeled invasive data, PET (positron emission tomography), and MRI (magnetic resonance imaging) data are used to train machine learning models and make predictions. The output is the result of predicting whether the patient is Pres., MCI, or AD. These results can be used for the early diagnosis of AD. Furthermore, to develop accurate early diagnosis techniques, it is crucial to improve machine-learning models with feedback from diagnostic data.

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
