# Peer review of "Biomarkers for Alzheimer’s Disease in the Current State: A Narrative Review"

_ijms, 2022, doi:10.3390/ijms23094962_

Round 1
Reviewer 1 Report
This manuscript is reviewing the literature on the current state of AD biomarkers which are in use to diagnose specific stages of AD and predict risk assessment of developing MCI or AD. Ideally more markers specifying the presymptomatic stage of the disease should be identified to predict risk assessment of developing the disease; in addition, ML/AI-based biomarkers coupled to the use of multiple non-invasive biomarkers should be further elucidated to tackle AD prevention. While the review is in general well-written, it also misses clarity with few sentences not complete and the lack of subheadings to better guide the reader throughout the (story that the authors want to tell with their) review. Therefore I do have some minor revisions:
** Line 31 - remove “silver tsunami” - this metaphor gives a negative connotation about the ageing population which in the context of AD may appear inappropriate.
** Figure 1 is not clear. pTau is not specified in the legend of the figure. Authors stated in the text that MCI is characterized by mild cognitive deficits, by definition, but in the figure the cognitive impairments only appear in the dementia stage. This provides confusion due to a lack of consistency between the main text and the figure/legend.
** Correct typos throughout the whole manuscript
** Section 3 - Add subheadings of conventional biomarkers to guide the reader through their description more easily
** Section 3 - Please explain what CDR 0.5, CDR 1, and CDR 2 stands for?
** Add a summary and transition statement at the end of section 3 - ideally at the end of section 2 as well - to better guide the reader through (the story you want to tell with) your review. Connection links between sections with transition/summary statements are missing.
** Line 185. Specify the abbreviation VP and explains what DP and VP stand for, related to the MCI stages - this is not clear and comes unexpectedly when things are rather well explained above. This section introduces several new terms and notions never explained in section 2 for instance or at the beginning of section 4. Please clarify and make it smoother/more linear to read.
** Line 186. CN is written; did the authors mean NC for cognitively normal (CN) but then see line 185 “(NC)”? Please verify typo and abbreviations throughout the ms to avoid confusion of the reader. Please also review the statements line 184-186 as this lacks clarity and rather introduces confusion - what the 2-metabolite panel versus 3-metabolite panel discriminate in terms of AD stages?
** Could the authors specify what means the value of the AUC in the context of section 4?
** Section 4 - Add subheadings to make it more linear to read
** Line 202-203. Sentence is not complete - Please rephrase.
** Add a transition/summary statement at the end of section 4.
** Line 213 - typo. Add upper letter AI.
** Line 243-244. Sentence not complete thus lacks clarity.
** Review the legend of Figure 3, e.g., lines 251-252. Few sentences not complete thus lack clarity and makes understanding difficult. A precise description of the figure should be provided.
** A final statement is missing to close section 5.
** Conclusion: what is ATN? This was not mentioned before.
Reviewer 2 Report
In this article the authors aim to describe an update on the non-invasive or minimally invasive techniques for identifying biomarkers. In addition, the topic includes AD diagnostic tests that enable mass screening to predict the risk of AD in the early and mild phases. This review also consists of the recent developments in ML employed in AD diagnosis that can be used along with the biomarkers to improve the accuracy of diagnosis and the subject differentiation capacity.
There are however some points to consider which I think will improve the understanding and coherence of this manuscript:
- Please correct the typos errors. This will apply to the whole manuscript.
- It would be interesting to correlate the information in the sections. Each section should be enriched with more recent data. The data presented in this version do not bring anything new.
- It would be interesting to present the most common combinations of biomarkers, those that can provide the most information related to clinical signs
- The conclusions are very general and I would expect them to be more to the point.
Author Response
We thank the reviewers for their careful reading of our manuscript and for their valuable, constructive feedback. We carefully read the reviewers’ comments and revised our manuscript accordingly.
Our responses to the reviewers are as follows.
Reviewer 2
Comments and Suggestions for Authors
In this article the authors aim to describe an update on the non-invasive or minimally invasive techniques for identifying biomarkers. In addition, the topic includes AD diagnostic tests that enable mass screening to predict the risk of AD in the early and mild phases. This review also consists of the recent developments in ML employed in AD diagnosis that can be used along with the biomarkers to improve the accuracy of diagnosis and the subject differentiation capacity.
There are however some points to consider which I think will improve the understanding and coherence of this manuscript:
Please correct the typos errors. This will apply to the whole manuscript.
We carefully revised the manuscript and used an English proofreading service to correct our manuscript further.
It would be interesting to correlate the information in the sections. Each section should be enriched with more recent data. The data presented in this version do not bring anything new.
It would be interesting to present the most common combinations of biomarkers, those that can provide the most information related to clinical signs
The conclusions are very general and I would expect them to be more to the point.
Thank you for your suggestions. We added references (ref 44, 50) and added text to each chapter. We also revised the concluding sentences (Lines 288-289). New sections are indicated by yellow highlighting.
Round 2
Reviewer 2 Report
I am satisfied with the answers and now your enriched manuscript looks very good.